# Association between Uranium Exposure and Thyroid Health: A National Health and Nutrition Examination Survey Analysis and Ecological Study

**DOI:** 10.3390/ijerph17030712

**Published:** 2020-01-22

**Authors:** Maaike van Gerwen, Naomi Alpert, Wil Lieberman-Cribbin, Peter Cooke, Kimia Ziadkhanpour, Bian Liu, Eric Genden

**Affiliations:** 1Department of Otolaryngology-Head and Neck Surgery, Icahn School of Medicine at Mount Sinai, New York, NY 10029, USA; peter.cooke@icahn.mssm.edu (P.C.); kimia.ziadkhanpour@icahn.mssm.edu (K.Z.); Eric.Genden@mountsinai.org (E.G.); 2Institute for Translational Epidemiology, Icahn School of Medicine at Mount Sinai, New York, NY 10029, USA; Naomi.alpert@mountsinai.org (N.A.); wil.lieberman-cribbin@icahn.mssm.edu (W.L.-C.); 3Department of Population Health Science and Policy, Icahn School of Medicine at Mount Sinai, New York, NY 10029, USA; bian.liu@mountsinai.org

**Keywords:** environmental exposure, thyroid health, radiation, uranium

## Abstract

Besides specific, incidental radiation exposure, which has been associated with increased thyroid cancer risk, the effects of exposure to background radiation from uranium, a naturally occurring, radioactive, and ubiquitous element, on the thyroid gland has not been widely studied. We therefore investigated the association between uranium exposure and thyroid health in the US. Using the National Health and Nutrition Examination Survey (NHANES), we assessed the association between urinary uranium levels and thyroid-related antibodies, including thyroglobulin antibodies (TgAb) and anti-thyroid peroxidase (anti-TPO), in the general population. Secondly, we performed an ecological study of age-adjusted thyroid cancer incidence rates per state and sources of uranium exposure. We included 3125 eligible participants from the NHANES and found a significant association between increased TgAb and increased urinary uranium levels when analyzed as quartiles (*p* = 0.0105), while no association was found with anti-TPO. In addition, although no significant correlation was found in the ecological study, certain states had high age-adjusted thyroid cancer incidence rates and a high number of uranium activity locations and high uranium concentrations in water. The present study suggests that uranium exposure may affect thyroid health, which warrants increased sampling of soil and water in high-risk states.

## 1. Introduction

The effect of radiation exposure on the thyroid gland has been extensively studied [1]; radiation, including exposure to nuclear accidents or incidents [2,3,4,5,6] and radiation therapy in childhood [7,8,9] has been associated with an increased risk of both thyroid cancer as well as benign thyroid nodules. [10]. Aside from these specific exposure scenarios, the general population is also exposed to background radiation from uranium, which is a naturally occurring, radioactive element with a wide distribution in the soil and higher concentrations in certain rock formations, especially within North America, Africa and Australia [11]. Furthermore, small amounts of uranium have been found in virtually all plants, animals and aquifers [11]. Uranium has 17 known isotopes but only ^234^U, ^235^U, and ^238^U are found in the environment due to the sufficient long half-lives of ^235^U and ^238^U; ^234^U is present as the decay product of ^238^U [11]. All uranium isotopes are known to be radioactive causing radiotoxicity due to alpha-particle emissions [11]. Additionally, the chemical toxicity of uranium may be associated with negative effects of certain uranium species, particularly in the kidneys. The chemical toxicity is therefore currently the limiting factor determining the current US Environmental Protection Agency (EPA) maximum allowed contaminant level of 30 µg/L in drinking water [12]. The association between the exposure to environmental background uranium and thyroid health problems is less clear.

Uranium is released into the environment through wind and water erosion, as well as mining, milling or other forms of uranium processing [13]. Uranium concentrations exceeding the EPA maximum allowed contaminant level were found in groundwater samples from two major aquifers in the US, High Plains and Central Valley [14]. Furthermore, increased uranium concentrations were found in groundwater, which is tapped for public water supply associated with increased irrigation and pumping of groundwater, combined with agricultural and urban development in California [15]. Volcanic eruptions are also associated with the release of uranium into the environment, which is shown by increased urinary uranium concentrations of residents of these regions [16].

Uranium can be absorbed into the body through ingestion of contaminated food or drink or inhalation of uranium-containing dust particles or aerosols. Upon absorption, uranium enters the blood and is predominantly deposited in kidney and bone or excreted in the urine [11]. Up to two-thirds of the absorbed uranium is found in bones, where it can remain for a long time with a biological half-life of 70 to 200 days due to biological clearance [13]. Absorbed uranium leaves the body in urine, making urinary uranium concentrations an indicator of exposure [17].

Some previous studies reported on a possible association between environmental uranium exposure and thyroid health problems. Increased thyroid cancer incidence has been documented in volcanic regions where increased urinary uranium concentrations have been found, although the underlying mechanism is still not clearly understood [16,18,19]. Furthermore, chronic uranium exposure has been associated with autoimmune thyroiditis in rats [20], while elevated incidence rates of thyroid diseases seem to occur in areas with increased uranium concentrations in drinking water [21].

Thyroid health problems may be indicated by increased levels of thyroid-related antibodies, including thyroglobulin antibodies (TgAb) and anti-thyroid peroxidase (anti-TPO). The presence of these antibodies has been associated with a diagnosis of autoimmune disease of the thyroid [22,23], although anti-TPO is a more specific marker for autoimmune thyroiditis than TgAb [24]. Kim et al. found that while anti-TPO was not associated with thyroid malignancy, positive TgAb could predict thyroid cancer in patients with thyroid nodules [24]. The increased tumor cell mass in patients with thyroid cancer releasing more thyroglobulin may stimulate chronic immunologic responses therefore producing more TgAb which would explain the association between positive TgAb and malignancy [24]. A meta-analysis confirmed that positive TgAb is an independent risk factor for thyroid cancer [25]. Investigating the association between urinary uranium as a marker of absorbed uranium and thyroid-related antibodies may provide further indications an increased risk of thyroid cancer associated with environmental uranium exposure.

Given that uranium is a ubiquitous, radioactive element, we proposed investigating the association between uranium exposure and thyroid health in the US, by (1) assessing the association between urinary uranium levels and thyroid-related antibodies, in particular TgAb, in the general population using the National Health and Nutrition Examination Survey (NHANES) and (2) performing an ecological study of thyroid cancer incidence and sources of uranium exposure.

## 2. Materials and Methods

### 2.1. NHANES Analysis

#### 2.1.1. Data Source and Study Population

The NHANES is an on-going cross-sectional survey of the health and nutrition status of the civilian, non-institutionalized population in the US, administered by the National Center for Health Statistics. To select a representative sample of the population, a complex, multi-stage probabilistic cluster design is used. The survey, conducted continuously since 1999, contains a questionnaire, exam, and laboratory information, with data released in a 2 year cycle. For this study, we selected NHANES participants from 2007 to 2010, who were part of the lab subsample with urinary uranium measurements (n = 5661). We limited the study to adult participants of 20 years and older, as information on thyroid disease and thyroid cancer was not available for those younger, with measures for anti-TPO and TgAb (n = 3572). Participants were excluded if they were taking thyroid medication (n = 248) as defined by Christensen et al. [26] and Turyk et al [27], if they ever had thyroid disease or thyroid cancer (n = 145), or if they were pregnant (n = 51). Those with outlier values of uranium exposure or thyroid antibodies (defined as high values at least double the next highest value) were also excluded (n = 3), resulting in a sample size of 3125 for this analysis (Figure 1).

#### 2.1.2. Outcomes and Covariates

The primary outcomes were anti-TPO and TgAb levels, and the primary predictor of interest was urinary uranium level. Reference ranges for anti-TPO and TgAb were taken from the NHANES laboratory documentation [28], with normal ranges for anti-TPO and TgAb of < 9 IU/mL and <4.0 IU/mL, respectively. Urine samples were analyzed using inductively coupled plasma-mass spectrometry to measure the total uranium concentration in urine [28]. In all years, the lower limit of detection for urinary uranium was 0.0017 μg/L. Values below that were imputed as 0.0017/,$\sqrt{2}$ consistent with NHANES practices [28]. Other covariates included gender, race/ethnicity, age in years, and body mass index (BMI).

#### 2.1.3. Statistical Analysis

The relationship between urinary uranium, anti-TPO and TgAb levels was examined using simple and multivariable linear regression models *(PROC SURVEYREG*) with uranium as both a continuous predictor, and divided into quartiles, similar to the approach taken by Christensen et al [26]. Uranium measures were creatinine adjusted (μg of uranium/g of creatinine) in all analyses and natural log (ln)-transformed in regression models where it was treated as a continuous predictor. Anti-TPO and TgAb values were also ln-transformed in regression models. Multivariable models were adjusted for gender, race/ethnicity, age, and BMI. *p*-values were obtained for each effect in the model, using the Wald’s *F* test in order to assess whether there was a significant association between urinary uranium (continuous or categorical) and thyroid antibodies, while controlling for covariates. Results were considered statistically significant if *p* < 0.05. All analyses were performed using SAS software, version 9.4 (SAS Institute, Cary, NC, USA). In order to account for the complex sampling strategy of the NHANES, the suite of survey procedures was used, and all analyses incorporated the survey design variables and weights. All statistics shown represent weighted values.

### 2.2. Ecological Study

#### 2.2.1. Thyroid Cancer Incidence

Statewide age-adjusted incidence rates (per 100,000 people) of thyroid cancer were obtained from the Centers for Disease Control and Prevention, and the National Cancer Institute United States cancer statistics for the year 2016 [29]. Rates were standardized to the 2000 U.S population and included all ages, races, and genders. These data were imported into ArcGIS to display statewide age-adjusted incidence rates of thyroid cancer for the contiguous United States (version 10.6.1; ESRI, Redlands, CA, USA).

#### 2.2.2. Uranium Measurements

Locations of uranium activity across the United States were obtained from the Uranium Location Database (ULD), a directory created to provide a spatial inventory of uranium mine and mill locations, claims, ore bodies, dumps, adits, ore-buying stations and mills to understand health and environmental impacts to nearby populations [30]. This database was compiled from multiple federal, state, and tribal government agency sources and contains approximately 15,000 uranium locations, including 4000 uranium-producing mines as of 2005.

Locations of uranium sampling of stream sediment, soil, groundwater and surface water were obtained from the National Uranium Resource Evaluation (NURE) Hydrogeochemical and Stream Sediment Reconnaissance database [31]. From 1975 to 1980, sampling was conducted to identify uranium resources throughout the United States. The NURE database contains 335,547 records of sampling locations, performed in all states except Iowa. Uranium concentrations were analyzed by delayed neutron counting, fluorescence spectroscopy, and mass spectrometry. According to the EPA, the maximum accepted contaminant level (MCL) for uranium in drinking water is 30 µg/L or 30 parts per billion (ppb), a guideline value shared by the World Health Organization [32].

#### 2.2.3. Statistical Analysis

Uranium sampling measurements in the NURE database with a concentration of ≥ 30 ppb and uranium activity locations were overlaid on the map of age-adjusted thyroid cancer incidence rate using ArcGIS.

The association between nuclear facilities (count per state), uranium activity locations (count per state; log transformed + 1), uranium concentration in water (average per state; log transformed + 1) and age-adjusted thyroid cancer incidence rates (log-transformed + 1 for the correlation analysis with uranium activity locations and uranium concentrations in water) were done using Spearman correlation. Analyses were performed using SAS software, version 9.4 (SAS Institute, Cary, NC, USA).

## 3. Results

### 3.1. NHANES Analysis

There were 3125 NHANES participants who met the selection criteria. The sample was majority male (53.8%) and non-Hispanic White (68.5%), with a mean age of 45.7 years, and mean BMI of 28.4 kg/m^2^. A majority of participants had TgAb and anti-TPO values within the normal range (93.8% and 89.9%, respectively). Of the sample, 9.1% had urinary uranium measurements below the limit of detection, and a median urinary uranium level of 0.006 μg/g creatinine (IQR: 0.003–0.011) (Table 1).

Univariate analysis of ln-transformed distributions of TgAb per quartile of creatinine-adjusted urinary uranium is shown in Figure 2. Mean TgAb concentrations increased across quartiles of urinary uranium, plateauing for quartile 4.

There was a statistically significant positive association between urinary uranium and ln TgAb when uranium was analyzed categorically (Wald’s *F* test *p* = 0.0105), although the effect appears to level off for the highest urinary uranium quartile. Results were similar when urinary uranium was treated as a continuous variable, although the association was not statistically significant (*p* = 0.0947) (Table 2). There was no significant association between urinary uranium and ln anti-TPO values in either model, although there was a significant correlation between ln TgAb and Ln anti-TPO (r = 0.44; *p* < 0.001).

### 3.2. Ecological Study

Age-adjusted thyroid cancer incidence rates per state ranged from 10.1–12 cases per 100,000 persons to 17.3–19.8 cases per 100,000 persons. Uranium activity locations were mainly located on the west side of the US while the nuclear facilities were mostly located on the east side. Sites with a uranium concentration of 30 µg/L (=30 ppb) or higher, which is above the EPA’s MCL, largely overlapped the uranium activity locations suggesting a correlation (Figure 3). Additional analyses showed a statistically significant correlation between uranium activity locations and uranium concentrations in water (β = 0.5781; *p* = 0.0002).

There was a potential cluster of elevated uranium concentrations in water, locations of uranium activity and higher age-adjusted thyroid cancer incidence (Figure 3). However, there was no statistically significant correlation between nuclear facilities and age-adjusted thyroid cancer incidence rates (r = −0.1432; *p* = 0.3161), uranium activity locations and age-adjusted thyroid cancer incidence rates (r = −0.1377; *p* = 0.3353) and uranium concentration in water and age-adjusted thyroid cancer incidence rates (r = 0.0657; *p* = 0.6608) (Figure 4). Certain states, specifically Wyoming, Utah, Colorado and North and South Dakota, do have high age-adjusted thyroid cancer incidence rates, a high number of uranium activity locations and high uranium concentrations in water, while Kansas has a high age-adjusted thyroid cancer incidence rate and high uranium concentration in water, without a high number of uranium activity locations (Figure 4).

## 4. Discussion

The present study on environmental uranium exposure and thyroid health is the first to combine a large database analysis and an ecological study. Thyroglobulin antibody levels were positively associated with urinary concentrations of uranium, suggesting that uranium exposure may affect thyroid health and potentially increase cancer risk as previous research has shown that TgAb positivity in patients with thyroid nodules was associated with an increased risk of suspicious cytology [33] and papillary thyroid cancer [34]. In addition, although we were unable to identify an overall significant correlation between sources of uranium exposures and increased thyroid cancer incidence rates, we identified a group of states where we reported high thyroid cancer incidence rates, an increased number of uranium activity locations and increased uranium concentrations in water, warranting increased surveillance of uranium exposure in the residents.

People are potentially exposed to uranium via ingestion of uranium-contaminated food or water after its release into the environment. Studies have shown that 41% of the ingested uranium by adults comes from beverages, 33% from vegetables and 26% from animal foodstuff [35]. Granite weathering soils produce uranium-rich vegetable forage and foods as well as uranium-rich mineral waters [35]. Studies that tested food and beverages reported that leafy vegetables, tea and herbs can be uranium-rich while fruits, seeds and flour are usually uranium-poor [35]. Furthermore, higher uranium concentrations are found in plants, especially leafy plants, in the immediate vicinity of uranium waste dumps compared to control areas [35]. Uranium concentrations evaluated in foodstuffs grown in agricultural soils located in former uranium mining areas also show high uranium concentrations in irrigation waters and some vegetables, including lettuce, potato, green bean pods and cabbage [36]. These findings suggest that residents of certain areas, namely those in regions surrounding granite-weathering soils, uranium waste dumps and former uranium mining areas, are at higher risk of environmental uranium exposure.

Sampling of soil and water on uranium is not performed on a regular basis in the US. The most recent data on uranium concentration in stream sediment, soil, groundwater and surface water came from sampling conducted between 1975 and 1980 from the NURE program initiated in 1973 [31]. This program’s initial goal was identifying uranium resources in the US and ended prematurely in 1980 due to lack of funding. To our knowledge, no new initiatives are currently undertaken to sample environmental uranium levels. Only one study on groundwater development on uranium in California showed that the median uranium concentrations increased over a 16 year time period [15]. The results of our ecological study based on the 1975–1980 data underline the importance of a renewal of the uranium sampling efforts to study the impact on public health, and to provide insight into temporal trends of uranium concentrations. This is especially important in states where high thyroid cancer incidence rates, high uranium concentrations in water and high numbers of uranium facilities are observed, such as Wyoming, Utah, Colorado, Kansas, and North and South Dakota. Wyoming, Colorado and Utah are Mountain States with rocky soil known to be rich of uranium, where mining is a prominent part of the economy [11].

When observing age-adjusted thyroid cancer incidence rates per state in the US, high rates in northeastern states stand out. The Three Mile Island incident with a partial nuclear reactor meltdown in Pennsylvania in 1979 was not found to be associated with increased thyroid cancer rates in the region surrounding the reactor [37]. To date, factors responsible for a more rapid increase in thyroid cancer incidence rates in Pennsylvania compared to the rest of the US remain unclear [38], although elevated thyroid cancer incidence rates in a contiguous area encompassing eastern Pennsylvania, New Jersey, and southern New York may indicate that local environmental factors play a role [38]. An etiological link between these elevated thyroid cancer incidence rates and exposure to radioactive iodine emissions from nuclear power reactors in this area was suggested by Mangano [39] but was not confirmed by a meta-analysis (Kim et al.) [40]. Future measurements of radiation as well as uranium concentrations in areas surrounding uranium mills and mines as well as nuclear facilities are needed to elucidate the possible association of an environmental exposure and thyroid cancer.

This study has several limitations: in order to investigate the association between urinary uranium levels and thyroid-related antibody concentrations, we were only able to include the 2007–2010 NHANES lab subsample, limiting the sample size. However, because the NHANES sampling is designed to include a representative sample of the total noninstitutionalized US population, we were able to include 3125 participants. Another important limitation is the cross-sectional nature of the data; since urinary uranium and thyroid-related antibodies were measured at the same time, it is not possible to determine a temporal association. Additionally, the NHANES does not record certain important variables associated with thyroid cancer risk, such as family history of thyroid cancer and history of genetic syndromes associated with increased thyroid cancer risk; we were therefore unable to adjust for these variables. The ecological study used data collected at different points in time ranging from 1975 to 1980 for the water uranium concentrations to 2016 for the cancer age-adjusted rates, thus a causal association is hard to be inferred. Furthermore, because of the ecological design of the study, and the aggregate data structure, no conclusions can be drawn at the individual level. Lastly, potential correlations may have been missed because of aggregation of data at state level.

Strengths of our study include the use of a nationally representative, population-based sample, rendering the results of the NHANES analysis generalizable to the general population. Additionally, we were able to consider many potential confounders such as demographic characteristics, use of thyroid medication, existence of thyroid cancer/ disease and pregnancy using this dataset. Although further exploration in prospective, longitudinal studies is needed, this current study is the first analysis investigating the association between environmental uranium exposure and thyroid health.

## 5. Conclusions

The present study suggests that uranium exposure may affect thyroid health, which warrants increased sampling of soil and water in states that showed high thyroid cancer incidences rates and an increased number of uranium activity locations and uranium concentrations in water, such as Wyoming, Utah, Colorado, Kansas and North and South Dakota. This is even more important as these states are large contributors to food production, therefore potentially exposing a large population. Future, preferably longitudinal, studies are needed to investigate the association between environmental uranium exposure and the risk of thyroid cancer or other thyroid-related health effects.

## Figures and Tables

**Figure 1 ijerph-17-00712-f001:**
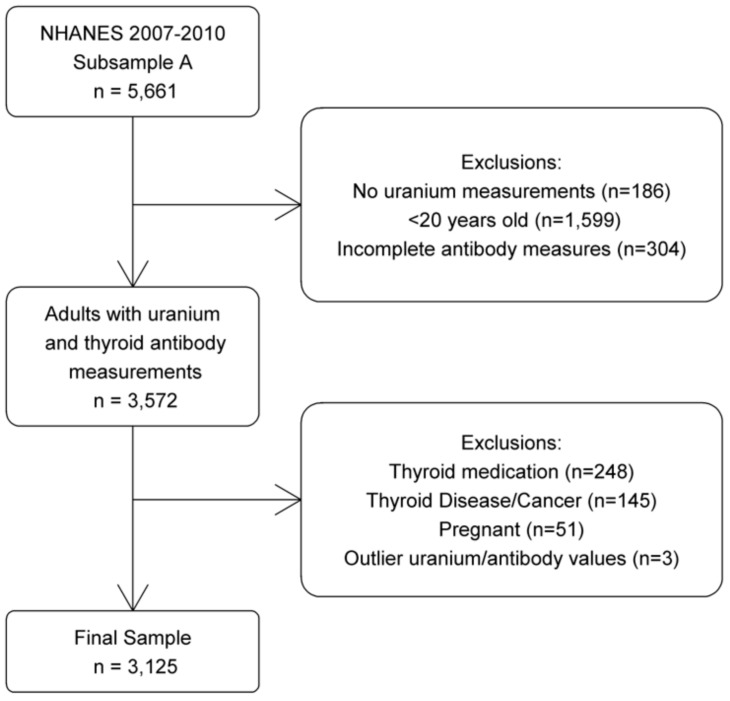
Selection criteria.

**Figure 2 ijerph-17-00712-f002:**
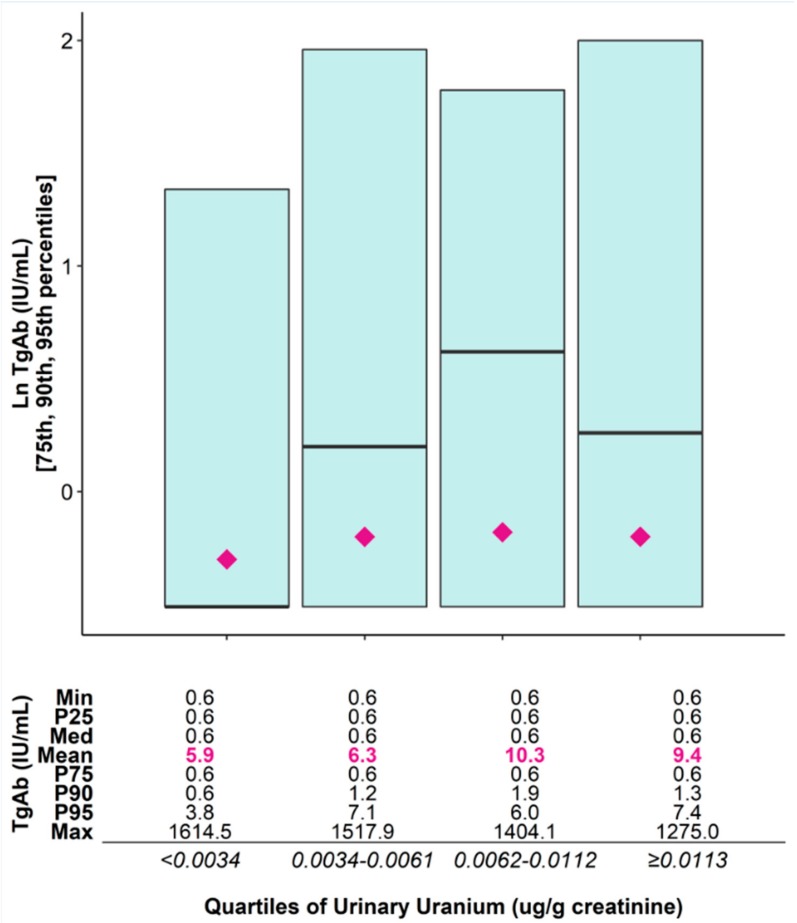
Ln-transformed serum concentration of thyroglobulin antibodies (TgAb) (IU/mL) per quartile of creatinine-adjusted urinary uranium (µg/g creatinine).

**Figure 3 ijerph-17-00712-f003:**
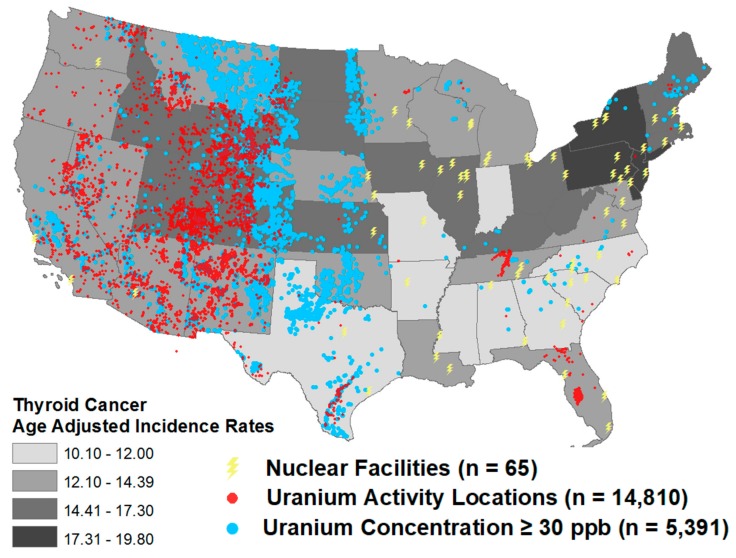
Distributions of age-adjusted thyroid cancer incidence rates per state and sources of uranium exposure, including uranium concentrations in water.

**Figure 4 ijerph-17-00712-f004:**
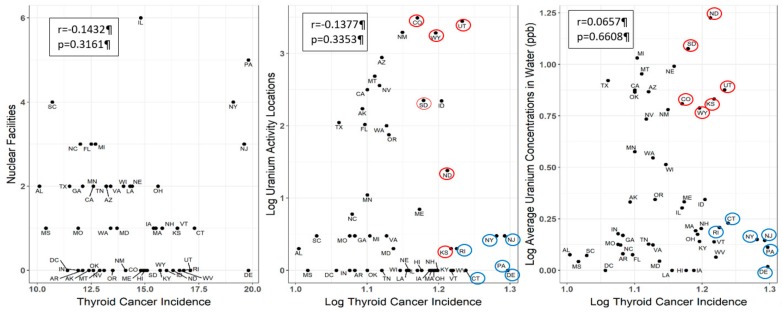
Spearman correlation of age-adjusted thyroid cancer incidence rates per state and sources of uranium exposure, including nuclear facilities, uranium activity locations and uranium concentrations in water. Note: The red-circled states are the states with high age-adjusted thyroid cancer incidence rates and high number of uranium activity locations and high uranium concentrations in water. The blue-circled states are the states with the highest age-adjusted thyroid cancer incidence rates in the US.

**Table 1 ijerph-17-00712-t001:** Description of the National Health and Nutrition Examination Survey (NHANES) population under study (n = 3125).

Variable		
Demographics		
	Mean (SE)	
Age (years)	45.7 (0.5)	
Body Mass Index (kg/m^2^)	28.4 (0.1)	
Gender	**n (%)**	
Male	1691 (53.8)	
Female	1434 (46.2)	
Race/Ethnicity		
Non-Hispanic White	1443 (68.5)	
Non-Hispanic Black	611 (11.3)	
Hispanic/Other	1071 (20.2)	
Thyroid Antibodies		
TgAb		
<4.0 IU/mL	2942 (93.8)	
≥4.0 IU/mL	183 (6.2)	
Anti-TPO		
<9.0 IU/mL	2839 (89.9)	
≥9.0 IU/mL	286 (10.1)	
	Median (IQR)	
TgAb (IU/mL)	0.60 (0.60–0.60)	
Anti-TPO (IU/mL)	0.62 (0.31–1.54)	
Uranium	Median (IQR)	% below LOD
Urinary Uranium (ug/g creatinine) *	0.006 (0.003–0.011)	9.1%

* Creatinine-adjusted. Abbreviations: TgAb, thyroglobulin antibodies; TPO, thyroid peroxidase; IQR, Interquartile Range; IU, international unit; LOD, limit of detection.

**Table 2 ijerph-17-00712-t002:** Adjusted * beta coefficients (95% CI) for the association between urinary uranium and thyroid antibodies.

Model	Ln TgAb (IU/mL)		Ln Anti-TPO (IU/mL)	
	Β_adj_ * (95% CI)	*p*-Value	Β_adj_ * (95% CI)	*p*-Value
Continuous				
Ln Uranium (ug/g creatinine)	0.035 (−0.006; 0.077)	0.0947	0.035 (−0.060; 0.130)	0.4569
Quartile (ug/g creatinine)				
<0.0034	Ref	0.0105	Ref	0.7552
0.0034–0.0061	0.110 (0.021; 0.198)		0.117 (−0.110; 0.343)	
0.0062–0.0112	0.114 (0.010; 0.218)		0.031 (−0.236; 0.298)	
≥0.0113	0.077 (−0.017;0.171)		0.046 (−0.188;0.281)	

Abbreviations: TgAb: thyroglobulin antibodies; TPO: thyroid peroxidase: CI, confidence interval * Adjusted for age, gender, race/ethnicity, and body mass index, n = 3092.

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
