# Peer review of "Association between Uranium Exposure and Thyroid Health: A National Health and Nutrition Examination Survey Analysis and Ecological Study"

_ijerph, 2020, doi:10.3390/ijerph17030712_

Round 1

Reviewer 1 Report

In this well-structured paper, authors investigated the association between urinary uranium levels and thyroid-related antibodies in the general population. Also, they performed Spearman correlation test between age adjusted thyroid cancer incidence rates per state and sources of uranium exposure using ecological study design. The authors found a significant association between increased TgAb and increased urinary uranium levels when analyzed as quartiles. The limitation of ecological study design is acknowledged in the manuscript.

I have several comments and suggestion:

I suggest authors to add natural uranium isotopic composition in the introduction part of the manuscript. That would better describe radiological hazard of natural uranium. In addition, based on ratios of activity concentration of specific radionuclides, it is possible to determine whether the measured sample contains natural of depleted uranium which have different specific activities (Bq/mg) leading to different absorbed doses and subsequent health effects. Please make distinction in the introduction between radiological and chemical toxicity of uranium due to the fact that radiotoxicity and chemotoxicity includes different biological mechanisms. In fact, EPA and WHO limit of 30 µg/l refers to kidney chemical toxicity of uranium. Detail description is provided in UNSCEAR Report, Annex D: BIOLOGICAL EFFECTS OF SELECTED INTERNAL EMITTERS—URANIUM, in Sources, Effects and Risks of Ionizing Radiation: UNSCEAR 2016 Report to the General Assembly, Scientific Annexes A, B, C and D, page 370. Please specify in the text that 70 to 200 days refers to biological half-life due to biological clearance rather that physical half-life due to radioactive decay (line 56). Please add a paragraph in the Methods section with description of methods used for Uranium measurement in urine samples. Did you measure total Uranium content? If so, please specify it. I suggest authors to use different marks to separate 95% lower and upper bounds in Table 2. (commas or semicolons instead of hyphen).

Author Response

Dear reviewer,

Thank you for your comments and suggestions. You will find our response to your comments below: 

I suggest authors to add natural uranium isotopic composition in the introduction part of the manuscript. That would better describe radiological hazard of natural uranium.

Thank you for this suggestion. I have added this information to the first paragraph of the introduction as suggested: “Uranium has 17 known isotopes but only 234U, 235U, and 238U are found in the environment due to the sufficient long half-lives of 235U and 238U; 234U is present as the decay product of 238U. [11] All uranium isotopes are known to be radioactive causing radiotoxicity due to alpha-particle emissions. [11]”

 In addition, based on ratios of activity concentration of specific radionuclides, it is possible to determine whether the measured sample contains natural of depleted uranium which have different specific activities (Bq/mg) leading to different absorbed doses and subsequent health effects. Please make distinction in the introduction between radiological and chemical toxicity of uranium due to the fact that radiotoxicity and chemotoxicity includes different biological mechanisms. In fact, EPA and WHO limit of 30 µg/l refers to kidney chemical toxicity of uranium. Detail description is provided in UNSCEAR Report, Annex D: BIOLOGICAL EFFECTS OF SELECTED INTERNAL EMITTERS—URANIUM, in Sources, Effects and Risks of Ionizing Radiation: UNSCEAR 2016 Report to the General Assembly, Scientific Annexes A, B, C and D, page 370.

I have added this important information to the introduction referencing the UNSCEAR 2016 report: “Additionally, the chemical toxicity of uranium may be associated with negative effects of certain uranium species, particularly in the kidneys. The chemical toxicity is therefore currently the limiting factor determining the current US Environmental Protection Agency (EPA) maximum allowed contaminant level of 30 µg/L in drinking water. [12]”

Please specify in the text that 70 to 200 days refers to biological half-life due to biological clearance rather that physical half-life due to radioactive decay (line 56).

I have clarified this: “ Up to two-third of the absorbed uranium is found in bones where it can remain for a long time with a biological half-life of 70 to 200 days due to biological clearance.”

Please add a paragraph in the Methods section with description of methods used for Uranium measurement in urine samples. Did you measure total Uranium content? If so, please specify it.

I have added information on the methods used by NHANES to measure uranium to the paragraph “outcomes and covariates “ of the methods section: “Urine samples were analyzed using inductively coupled plasma-mass spectrometry to measure the total uranium concentration in urine. [26] In all years, the lower limit of detection for urinary uranium was 0.0017 μg/L.  Values below that were imputed as 0.0017/, consistent with NHANES practices.”

NHANES measured the total uranium concentration.

I suggest authors to use different marks to separate 95% lower and upper bounds in Table 2. (commas or semicolons instead of hyphen). 

I have changed hyphens to  semicolons in table 2 as suggested.

Reviewer 2 Report

The authors propose to assess the relationship between environmental uranium exposure and thyroid health, particularly the risk of thyroid cancer, in two distinct approaches:

In the first part, the relationship between urinary uranium concentrations and TgAB and TPO are studied in 3125 subjects.

Clarification should be provided to justify a focus on these 2 markers:

- It was suggested in an earlier publication that TgAB is a marker of thyroid cancer in patients with thyroid nodule (ref 23). What are the known or hypothesized mechanisms to explain the association between TgAB and thyroid malignancy ? Was this association confirmed in independent studies ?

- p.6 line 113-114: I don't understand the meaning of “anti-TPO was interpreted as negative control”. Please clarify

- What is the underlying hypothesis for testing the association between urinary uranium and TgAB and TPO? If an association is observed, can a causal link be inferred between environmental exposure to uranium and thyroid cancer risk? The hypothesis should be clearly stated.

Improvements should be made in the presentation and interpretation of results:

- Figure 2 is not clear. Box plots showing the entire TgAB distribution in each quartile should be provided.

- Table 2: what is the p-value associated with uranium quartiles? A test for trend? How was it calculated? Please specify in the M&M section.

- The urinary uranium - TgAB relationship does not increase monotonically. Thus the p-value at 0.0105 is questionable. These results should not be over-interpreted as a strong positive finding.

In the second part:

- the correlation analysis between standardized incidence rates and uranium exposure indicators is performed at the US State level. At this macroscopic scale, strong ecological biases may occur and results must be interpreted with extreme caution.  The conclusion pointing to the States with both high incidence rates and high activity location or urinary water concentrations is  not based on substantiated statistical evidence.

- As a perspective, if the data are available, I would suggest to conduct a spatial analysis at a finer level (i.e. county?), using models taking into account the geolocalised sources of exposure and the incidence of thyroid cancers measured by age, sex, and year of diagnosis.

Author Response

Dear reviewer,

Thank you for your comments and suggestions. You will find our responses below: 

In the first part, the relationship between urinary uranium concentrations and TgAB and TPO are studied in 3125 subjects.

Clarification should be provided to justify a focus on these 2 markers:

- It was suggested in an earlier publication that TgAB is a marker of thyroid cancer in patients with thyroid nodule (ref 23). What are the known or hypothesized mechanisms to explain the association between TgAB and thyroid malignancy ? Was this association confirmed in independent studies ?

Thank you for this suggestion. I have elaborated and edited the paragraph in the introduction explaining the focus on these 2 markers:” Thyroid health problems may be indicated by increased levels of thyroid-related antibodies, including thyroglobulin antibodies (TgAb) and anti-thyroid peroxidase (anti-TPO).  The presence of these antibodies has been associated with a diagnosis of autoimmune disease of the thyroid [22, 23], although anti-TPO is a more specific marker for autoimmune thyroiditis than TgAb. [24] Kim et al found that while anti-TPO was not associated with thyroid malignancy, positive TgAb could predict thyroid cancer in patients with thyroid nodules. [24]  The increased tumor cell mass in patients with thyroid cancer releasing more thyroglobulin may stimulate chronic immunologic responses therefore producing more TgAb which would explain the association between positive TgAb and malignancy. [24] A meta-analysis confirmed that positive TgAb is an independent risk factor for thyroid cancer. [25]”

I also added the reference of a meta-analysis confirming the association between TgAb and thyroid malignancy.

- p.6 line 113-114: I don't understand the meaning of “anti-TPO was interpreted as negative control”. Please clarify

Anti-TPO is referred to as negative control because, as described previously, anti-TPO is not associated with thyroid cancer while positive TgAb is.

The following phrase has been added: “TgAb was considered a marker for thyroid cancer risk while anti-TPO was interpreted as negative control it has not been associated with increased thyroid cancer risk. [24, 25]

- What is the underlying hypothesis for testing the association between urinary uranium and TgAB and TPO? If an association is observed, can a causal link be inferred between environmental exposure to uranium and thyroid cancer risk? The hypothesis should be clearly stated.

I have added the following sentence to the introduction to state this hypothesis: “Investigating the association between urinary uranium as a marker of absorbed uranium and thyroid-related antibodies may provide further indications an increased risk of thyroid cancer associated with environmental uranium exposure.”

Improvements should be made in the presentation and interpretation of results:

- Figure 2 is not clear. Box plots showing the entire TgAB distribution in each quartile should be provided.

We decided to submit the figure showing the mean, 75th, 90th and 95th percentiles because box plots showing the entire distribution of TgAb, including the maximum, was not informative (this figure can be provided if needed). We therefore decided to provide all the data beneath the boxplots.

- Table 2: what is the p-value associated with uranium quartiles? A test for trend? How was it calculated? Please specify in the M&M section.

P-values were obtained from the linear regression models.  There is one p-value for the quartiles because we were looking at the p-value for the test of model effects, which looks at the variable as a whole. 

We have updated the Materials and Methods section as follows: “The relationship between urinary uranium, anti-TPO and TgAb levels was examined using simple and multivariable linear regression models (PROC SURVEYREG) with uranium as both a continuous predictor, and divided into quartiles, similar to the approach taken by Christensen et al. [26] Uranium measures were creatinine adjusted (μg of uranium/g of creatinine) in all analyses and natural log (ln)-transformed in regression models where it was treated as a continuous predictor. Anti-TPO and TgAb values were also ln-transformed in regression models. Multivariable models were adjusted for gender, race/ethnicity, age, and BMI. P-values were obtained for each effect in the model, using the Wald’s F test in order to assess whether there was a significant association between urinary uranium (continuous or categorical) and thyroid antibodies, while controlling for covariates.  Results were considered statistically significant if p<0.05.

- The urinary uranium - TgAB relationship does not increase monotonically. Thus the p-value at 0.0105 is questionable. These results should not be over-interpreted as a strong positive finding.

We see an increase for Q2 vs Q1 and a further increase for Q3 vs Q1, but the point estimate for Q4 vs Q1 is a little lower than the previous. We seem to have a little bit of a drop off in TgAb for the highest values of urinary uranium, which may indicate that results level off at a certain point.

We have updated the results as follows: “There was a trend towards increased TgAb with increased urinary uranium values when uranium was treated as continuous variable (p=0.0947), and a statistically significant trend when uranium was analyzed as quartiles (p=0.0105), which seems to level off for the highest urinary uranium quartile.”

In the second part:

- the correlation analysis between standardized incidence rates and uranium exposure indicators is performed at the US State level. At this macroscopic scale, strong ecological biases may occur and results must be interpreted with extreme caution.  The conclusion pointing to the States with both high incidence rates and high activity location or urinary water concentrations is not based on substantiated statistical evidence.

Thank you for your suggestion. I have edited the paragraph in the results section: “Certain states, specifically Wyoming, Utah, Colorado and North and South Dakota, do have high age-adjusted thyroid cancer incidence rates, a high number of uranium activity locations and high uranium concentrations in water, while  Kansas has a high age adjusted thyroid cancer incidence rate and high uranium concentration in water, without a high number of uranium activity locations. (Figure 4)

- As a perspective, if the data are available, I would suggest to conduct a spatial analysis at a finer level (i.e. county?), using models taking into account the geolocalised sources of exposure and the incidence of thyroid cancers measured by age, sex, and year of diagnosis.

Although we have counts of the uranium activity locations, nuclear facilities and concentrations of uranium in water at county level, the age adjusted thyroid cancer incidence rate per county for certain states including counties in the states of interest is not available due too low number of people per county. Furthermore, certain states due not release data on county level because of state legislation and regulations. We therefore decided to only analyze at state level.

Reviewer 3 Report

This study analyzed the effect of uranium exposure and antibody against thyroglobulin. A second aim was to perform an ecological study of thyroid cancer incidence and sources of uranium exposure. Data comes from NHANES.

There are several points that need to be clarified.

Line 90: if the aim of the article is “performing an ecological study of thyroid cancer incidence and sources of uranium exposure”, why subjects with thyroid cancer have been excluded? Clarify

Line 113: why TPO has been considered as a negative control?

Figure 2 and related analyses: did the authors include only subjects with positive tgab or all the sample? The reference range is low. Authors should report more information about the assay used for tgab. Did it discriminate, for example, between 3 or 4 UI/ml?

Figure 2 also showed that 25 and 50 centile are the same for each quartile of uranium

If not, analyses should be limited to subjects with tgab positivity. What if tgab are tested as dichotomic variable?

Author Response

Dear reviewer,

Thank you for your comments and suggestions. You will find our responses below:

This study analyzed the effect of uranium exposure and antibody against thyroglobulin. A second aim was to perform an ecological study of thyroid cancer incidence and sources of uranium exposure. Data comes from NHANES.

There are several points that need to be clarified.

Line 90: if the aim of the article is “performing an ecological study of thyroid cancer incidence and sources of uranium exposure”, why subjects with thyroid cancer have been excluded? Clarify

Thank you for your comments. For the first part of the study (the NHANES analysis), patients with thyroid cancer were excluded because this may have given a false positive association because patients with thyroid cancer may have increased thyroid-related antibodies. The ecological part of the study, which is the second part of the study, included thyroid cancer incidence.

Line 113: why TPO has been considered as a negative control?

Anti-TPO is referred to as negative control because anti-TPO is not associated with thyroid cancer while positive TgAb is. We have edited the introduction to clarify this:” Thyroid health problems may be indicated by increased levels of thyroid-related antibodies, including thyroglobulin antibodies (TgAb) and anti-thyroid peroxidase (anti-TPO).  The presence of these antibodies has been associated with a diagnosis of autoimmune disease of the thyroid [22, 23], although anti-TPO is a more specific marker for autoimmune thyroiditis than TgAb. [24] Kim et al found that while anti-TPO was not associated with thyroid malignancy, positive TgAb could predict thyroid cancer in patients with thyroid nodules. [24]  The increased tumor cell mass in patients with thyroid cancer releasing more thyroglobulin may stimulate chronic immunologic responses therefore producing more TgAb which would explain the association between positive TgAb and malignancy. [24] A meta-analysis confirmed that positive TgAb is an independent risk factor for thyroid cancer. [25]”

The following phrase has been added to the methods section: “TgAb was considered a marker for thyroid cancer risk while anti-TPO was interpreted as negative control it has not been associated with increased thyroid cancer risk. [24, 25]

Figure 2 and related analyses: did the authors include only subjects with positive tgab or all the sample? The reference range is low. Authors should report more information about the assay used for tgab. Did it discriminate, for example, between 3 or 4 UI/ml?

NHANES data file contains continuous values for both thyroid-related antibodies including 1 decimal; it is therefore possible to discriminate between 3 and 4 UI/ml. All participants are included in the analysis, including participants with a value lower than the normal values of these antibodies.

Figure 2 also showed that 25 and 50 centile are the same for each quartile of uranium

This is most likely associated with the fact that most people have thyroid antibody concentrations within the normal range as shown in table 1: 93.8% of the population has a TgAb concentration of <4.0 IU/mL and 89.9% of the population has anti-TPO concentration of <9.0 IU/mL.

If not, analyses should be limited to subjects with tgab positivity. What if tgab are tested as dichotomic variable?

NHANES data file contains continuous values for both thyroid-related antibodies including 1 decimal; it is therefore possible to discriminate between 3 and 4 UI/ml. All participants are included in the analysis, including participants with a value lower than the normal values of these antibodies.

Round 2

Reviewer 2 Report

It is stated lines 202-203 "There was a trend towards increased TgAb with increased urinary uranium values when uranium was treated as continuous variable (p=0.0947), and a statistically significant trend when uranium was analyzed as quartiles (p=0.0105)..."

According to the MM section, the p=0.0105 is not derived from a trend test but from a linear model that does not take into account the increase in urinary uranium concentrations from Q1 to Q4. A trend test could be performed, e.g., by fitting a model using a continuous variable for uranium concentration equal to the median value in each quartile. It is therefore incorrect to say that there is a "statistically significant trend" .

In addition, the decrease of the beta values between Q3 and Q4 shows that no such trend exists. The whole statement is therefore misleading and should be reformulated. 

Author Response

Thank you for your comment and we understand that this may be confusing. We therefore reformulated the sentence as follows:

“There was a statistically significant positive association between urinary uranium and ln TgAb when uranium was analyzed categorically (Wald’s F test p=0.0105), although the effect appears to level off for the highest urinary uranium quartile. Results were similar when urinary uranium was treated as a continuous variable, although the association was not statistically significant   (p=0.0947). (Table 2)There was no significant association between urinary uranium and ln anti-TPO values in either model, although there was a significant correlation between ln TgAb and Ln anti-TPO (r= 0.44; p<0.001).”

Reviewer 3 Report

I do not agree with the choice of control group. I understand tpoab are not related with thyroid cancer. But both tgab and tpoab are commonly found in patients with chronic autoimmune thyroiditis and one cannot discriminate tgab related to uranium, exposure and tgab caused by thyroiditis (A further question is how did the authors manage those with both antibodies positivity?). A control group should be patients without both tgab an tpoab.

Author Response

Thank you for your comment and we acknowledge that the term negative control may be confusing. We have therefore deleted the following sentence from the Methods section: “TgAb was considered a marker for thyroid cancer risk while anti-TPO was interpreted as negative control it has not been associated with increased thyroid cancer risk. [24,25]”

For our analysis, we looked at anti-TPO and TgAb as two separate primary outcome measures of interest; we therefore did not separately investigate the participants with high values for both antibodies. No group was identified as a control group but we investigated the association between these two thyroid-related antibodies and urinary uranium levels.